# An Artificial Neural Network Model for the Prediction of Perioperative Blood Transfusion in Adult Spinal Deformity Surgery

**DOI:** 10.3390/jcm11154436

**Published:** 2022-07-29

**Authors:** Rafael De la Garza Ramos, Mousa K. Hamad, Jessica Ryvlin, Oscar Krol, Peter G. Passias, Mitchell S. Fourman, John H. Shin, Vijay Yanamadala, Yaroslav Gelfand, Saikiran Murthy, Reza Yassari

**Affiliations:** 1Spine Research Group, Montefiore Medical Center, Albert Einstein College of Medicine, New York, NY 10467, USA; mhamad@montefiore.org (M.K.H.); ygelfand@montefiore.org (Y.G.); samurthy@montefiore.org (S.M.); ryassari@montefiore.org (R.Y.); 2Department of Neurological Surgery, Montefiore Medical Center, Albert Einstein College of Medicine, New York, NY 10467, USA; jessica.ryvlin@einsteinmed.org; 3Division of Spinal Surgery, Department of Orthopaedic and Neurological Surgery, NYU School of Medicine, New York, NY 10467, USA; okrol90@gmail.com (O.K.); peter.passias@nyumc.org (P.G.P.); 4Spine Surgery Service, Department of Orthopaedic Surgery, Hospital for Special Surgery, New York, NY 10467, USA; fourmanm@hss.edu; 5Department of Neurosurgery, Harvard Medical School, Massachusetts General Hospital, Boston, MA 02114, USA; shin.john@mgh.harvard.edu; 6Hartford HealthCare, Westport, CT 06880, USA; vijay.yanamadala@hhchealth.org

**Keywords:** artificial intelligence, neural network, adult spinal deformity, transfusion, scoliosis

## Abstract

Prediction of blood transfusion after adult spinal deformity (ASD) surgery can identify at-risk patients and potentially reduce its utilization and the complications associated with it. The use of artificial neural networks (ANNs) offers the potential for high predictive capability. A total of 1173 patients who underwent surgery for ASD were identified in the 2017–2019 NSQIP databases. The data were split into 70% training and 30% testing cohorts. Eighteen patient and operative variables were used. The outcome variable was receiving RBC transfusion intraoperatively or within 72 h after surgery. The model was assessed by its sensitivity, positive predictive value, F1-score, accuracy (ACC), and area under the curve (AUROC). Average patient age was 56 years and 63% were female. Pelvic fixation was performed in 21.3% of patients and three-column osteotomies in 19.5% of cases. The transfusion rate was 50.0% (586/1173 patients). The best model showed an overall ACC of 81% and 77% on the training and testing data, respectively. On the testing data, the sensitivity was 80%, the positive predictive value 76%, and the F1-score was 78%. The AUROC was 0.84. ANNs may allow the identification of at-risk patients, potentially decrease the risk of transfusion via strategic planning, and improve resource allocation.

## 1. Introduction

Corrective surgery for adult spinal deformity (ASD) is associated with significant blood loss [1]. Perioperative red blood cell (RBC) transfusion rates have been estimated to range from 27% up to 90% in some series [2,3,4,5]. Although the use of blood products can improve tissue oxygenation, their administration does not come without health and economic costs. Multiple studies have shown associations between RBC transfusion and adverse events such as transfusion reactions, transfusion-related lung injury, infection, venous thromboembolism, and prolonged intensive care unit and overall length of stay [3,6,7,8,9].

Multiple studies have reported “risk factors” for transfusion in deformity surgery, but estimating an individual patient’s probability is challenging. Identifying a high-risk patient could allow for better preoperative optimization or alterations in the surgical plan to minimize the transfusion rate and associated complications [10]. In recent years, advanced computer algorithms such as artificial neural networks (ANN) have emerged as powerful tools for prediction in spine surgery [11,12,13,14]. These models are inspired by biological neural networks and are excellent at problems, such as classifying a patient into needing or not needing a transfusion based on input parameters.

In this study, we attempted to predict the need for RBC transfusion in ASD surgery by creating an ANN model. Input data from a prospectively collected database was used to train the model and a withheld dataset was used for testing.

## 2. Materials and Methods

### 2.1. Data Source and Patient Selection

This is a retrospective study of prospectively collected data. We utilized the American College of Surgeons National Surgical Quality Improvement Program (NSQIP) dataset from 2017 to 2019. The NSQIP prospectively collects patient and operative data on major surgical cases performed at over 700 participating institutions, as well as 30-day morbidity and mortality. A trained “Surgical Clinical Reviewer” at each institution performs data collection; current inter-rater reliability is over 95% [15].

Adult patients >18 years of age who underwent surgery for ASD were identified via the use of CPT codes (primary CPT code: 22206, 22207, 22800, 22802, 22804, 22808, 22810, 22812, 22818, 22819, 22843, 22844, or 22846). The initial search yielded 1446 patients; 273 were excluded (18.9%) given they had one or more of the following characteristics: missing data, emergency case, transfer from a facility other than home, disseminated cancer, outpatient surgery, ventilator dependency, preoperative sepsis, or a history of recent unintentional weight loss. The final analytic sample consisted of 1173 cases.

We followed the guidelines by Luo et al. for developing and reporting machine learning predictive models in biomedical research [16].

### 2.2. Collected Parameters

Collected data included patient age, sex, ASA (American Society of Anesthesiologists) class, smoking status, chronic steroid use, history of bleeding disorder, dependent functional status, body weight, preoperative hematocrit, surgeon specialty (neurosurgery vs. orthopedic surgery), surgery duration in hours, the use of pelvic fixation, the use of interbody grafts, the use of osteotomy or three-column osteotomy (3CO), the number of posterior levels fused (6–12 or 7–13), and revision status.

### 2.3. Model Creation

Our model was built to solve a diagnostic problem. It is a classification algorithm with the main outcome variable being the administration of at least one RBC unit intraoperatively or within 72 h after surgery (as defined by NSQIP). The 18 collected parameters were used as input neurons in the model based on previously reported risk factors for transfusion [5,17]. The feedforward ANN model was built using Python 3.0 (Python Software Foundation, Wilmington, DE, USA) and the TensorFlow (Google Brain, Mountain View, CA, USA) and Keras libraries (open source software, developed by Francois Chollet). For model creation, a 70/30 random split was made for training and testing data, respectively. Data were scaled before analysis. Ten percent of the data used for training was used for internal validation (150 epochs, verbose = 2). Several ANN models were tested with various hidden layers and different activation functions. These were compared based on the following accuracy metrics: sensitivity, positive predictive value, F1-score (harmonic mean between sensitivity and positive predictive value), and overall accuracy (ACC). The area under the curve of the receiver operating characteristic analysis (AUROC) was also calculated for the final model.

### 2.4. Data Analysis and Model Creation

Statistical analyses were performed in Stata IC 16 (StataCorp, College Station, TX, USA). Comparisons between no transfusion and transfusion groups were made via *t*-tests or chi-square tests when appropriate. Descriptive statistics were used for the study population. Statistical significance was defined as a probability (*p*-value) value less than 0.05.

## 3. Results

### 3.1. Patient Population

A total of 1173 cases were analyzed in this study (Table 1). Mean age was 55.7 years and 62.9% of patients were female. The most common ASA classification was 3 (56.8%), and 14.9% of patients were smokers. Average duration of surgery was 5.8 h and 3CO was performed in 19.5% of cases. Overall, the transfusion rate was 50.0% (586/1173 patients).

Comparisons between the no transfusion and transfusion groups are summarized in Table 2. Statistically significant differences were found across most parameters examined. Notably, patients requiring a transfusion were older (57 vs. 54 years, *p* = 0.005), had a higher comorbidity burden as determined by the ASA Class (*p* < 0.001), and they were also more likely to be dependent for activities of daily living (*p* < 0.001). Operative time was also significantly longer (7.1 vs. 4.4 h, *p* < 0.001) and these patients were more likely to undergo pelvic fixation (35.3% vs. 7.3%, *p* < 0.001), 3CO (28.3% vs. 10.7%, *p* < 0.001), and revision surgery (13.3% vs. 7.0%, *p* < 0.001), among other differences.

### 3.2. ANN Models

A summary of the different model architectures is shown in Table 3. They were all sequentially built on the training data (70% of patients) and compared via their accuracy metrics on the testing data (30% of patients). All models had 18 input neurons and 2–4 hidden layers (with 8–128 neurons per layer). Model validation was carried out on 10% of the data. Sigmoid or rectified linear unit (ReLU) activation functions were used. A dropout of 10–20% of neurons was also used to prevent overfitting.

After comparing the models, model #3 achieved the highest accuracy metrics. This model consisted of only two hidden layers (one with 128 and one with 64 neurons) and used ReLU activation function as the main function (Figure 1). The model achieved the highest sensitivity and positive predictive value of 0.80 and 0.76, respectively. Its F1-score was 0.78 and overall ACC was 0.77. The AUROC was calculated at 0.84.

## 4. Discussion

Corrective surgery for ASD is associated with high blood loss given the need for extensive soft tissue dissection, osteotomies, and prolonged operative duration [18]. Although RBC transfusion can improve tissue oxygenation and prevent hypoperfusion to the spinal cord and vital organs, multiple studies have reported an association between transfusion and postoperative complications [3,6,7,19,20,21,22,23]. Another current issue is the severe shortage of blood products resulting in a national crisis [24]. Thus, the implementation of predictive algorithms can potentially identify at-risk patients and reduce the potential transfusion rate or improve resource allocation. In this study, we created an ANN model in an attempt to identify at-risk patients for perioperative RBC transfusion. One of our models was able to correctly identify 80% of at-risk patients and showed a positive predictive value of 76% and an overall ACC of 77%.

The current estimated blood loss during deformity surgery is approximately 2 L [1]. Transfusion rates are high and although multiple studies have identified “risk factors” for their need [2,3], predicting their occurrence is based on a “best guess” by the clinician based on prior experience with patients presenting with similar risk factors. Puvanesarajah et al. examined 165 adult patients who underwent ASD surgery, finding an intraoperative transfusion rate of 90%. Authors found that preoperative hemoglobin < 11.5 g/dL, increasing operative time, and the use of osteotomies were all significantly associated with the number of allogenic RBC unites transfused intraoperatively [5]. In another investigation of 5805 patients, authors found age over 65 years, ASA Class ≥ 3, cardiac comorbidities, and bleeding disorders to all be independently associated with perioperative transfusion. In terms of surgical risk factors, independent predictors included posterior approaches, pelvic fixation, and osteotomy [17].

Compared to traditional statistical modeling, ANNs are considered deep-learning algorithms, a branch of artificial intelligence. These networks are inspired by biological neural networks and are very useful in classification problems. They use training data and can learn how much each input variable (such as age, sex, preoperative hematocrit, or use of osteotomies) influences a particular outcome; they are then able to adjust themselves accordingly to make the most accurate prediction possible. The model can then be tested on new data and its accuracy metrics calculated. Some of the most common neural networks include feedforward neural networks, recurrent neural networks, and convolutional neural networks. In feedforward networks such as the one used in this study, data travel from input neurons towards the output neuron(s). In recurrent networks such as Hopfield’s network, data/signals can travel in both directions and also within a same layer. Lastly, convolutional neural networks are a specialized algorithm predominantly used in image recognition.

When deciding which models to develop and test, no universal guidelines exist. However, some of the most common features that are adjusted include the number of hidden layers and activation functions. Hidden layers are layers of mathematical functions located between the input neurons (the patient/surgical parameters to be examined) and the output neurons (classifying a patient into no transfusion vs. transfusion). These layers contain a certain number of neurons/nodes that each produce a probability value from 0 to 1 depending on the activation function used. In terms of activation functions, these algorithms take the input from previously activated neurons and transforms the value into an ON or OFF state. Some of the most used activation functions include sigmoid and rectified linear unit functions.

Durand et al. also utilized advanced learning algorithms in an attempt to predict the need for RBC transfusion in ASD [2]. They used data on 824 patients for training and 205 patients for testing. Surgical duration, surgical invasiveness, preoperative hematocrit, and patient weight and age were the most influential parameters associated with transfusion [2]. Using a random forest classifier model, they achieved an AUROC of 0.85 (compared to 0.84 for our model). Unfortunately, other than the model’s discriminative capacity, no data were provided regarding its ACC, sensitivity, or positive predictive value on the testing data. Simply using the AUROC as a measurement of a model’s performance is insufficient, as it does not provide any clinically useful data [25].

Raman et al. performed a single-center review of 909 patients who underwent deformity correction and used decision-tree-based modeling to identify predictors of blood loss and transfusion [3]. They reported a 41.5% transfusion rate and found that the fusion of >13 levels, ASA Class > 1, hypertension, three-column osteotomies, pelvic fixation, and operative time > 8 h were all significant risk factors associated with perioperative transfusion [3]. Although they used a machine learning approach to identify risk factors, no predictive model was created.

On the other hand, our study’s proposed model can correctly predict 80% of patients needing a transfusion. When faced with an at-risk patient, several strategies could be employed such as the modification of the surgical plan to reduce the number of levels fused or osteotomies performed; improving intraoperative resuscitation without blood products; optimization of antifibrinolytic agents; preoperative iron supplementation; improving preoperative nutritional status; and others [10].

While our proposed model could correctly identify 80% of at-risk patients, one of the concerns that may arise is that an ANN is considered a “black box” model. This means that exact parameter weights are unknown and function optimization is carried out by the model without human input. In other words, we cannot know exactly which parameters (age, comorbidities, or surgical invasiveness for example) are the most “significantly associated” with transfusion. This machine learning philosophy differs from traditional statistical approaches that focus more on the specific variables associated with an outcome rather than focusing on the optimal predictive capability of the model.

Nonetheless, we believe the model can be useful as is given its high accuracy metrics. Further improvements in positive predictive value can be made (currently at 76%). This means that 24% of patients will be false positives and incorrectly classified as needing a transfusion. However, implementing blood-saving strategies would not necessarily be detrimental to patients. Although the NSQIP database contains high-quality data, information on curve types or degrees of deformity is missing and cannot be used as input parameters; thus, whether these data can improve the model’s performance is unknown. Another limitation is that NSQIP does not quantify the total units of RBCs given. The transfusion of one versus multiple units of blood reflects different patient profiles and circumstances; as such, it is unclear as to whether our model would perform differently when predicting single versus multiple transfusions. Not knowing the specific threshold for transfusion initiation is also a limiting factor. Thresholds of <7 g/dL or <8 g/dL may result in different transfusion rates and practices among surgeons or centers; thus, the model needs to be interpreted within each context. Ultimately, the development and implementation of predictive algorithms should be unique to each institution and target population.

## 5. Conclusions

Different ANN models were tested for their ability to predict perioperative RBC transfusion after surgery for ASD. One of the models achieved high sensitivity, being able to correctly classify 80% of patients needing a transfusion. Further research and external validation is needed. These advanced models may allow the identification of at-risk patients, potentially decrease the risk of transfusion via strategic planning, and improve resource allocation.

## Figures and Tables

**Figure 1 jcm-11-04436-f001:**
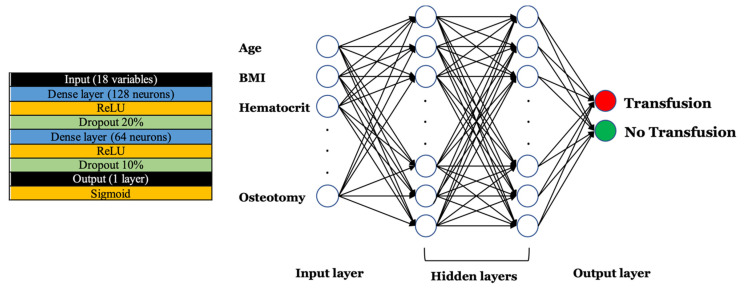
Artificial Neural Network architecture.

**Table 1 jcm-11-04436-t001:** Baseline and operative characteristics of all patients.

Parameter	Value
Age (mean, standard deviation)	55.7 (18.9)
Sex	
Male	435 (37.1%)
Female	738 (62.9%)
ASA Class	
1	56 (4.8%)
2	420 (35.8%)
3	666 (56.8%)
4	31 (2.6%)
Smoker	175 (14.9%)
Chronic steroid use	56 (4.8%)
Bleeding disorder	35 (3.0%)
Dependent functional status	62 (5.3%)
Body weight (mean kg, standard deviation)	77.8 (20.9)
Preoperative hematocrit (mean, standard deviation)	40.6 (4.5)
Orthopedic surgeon as attending	617 (52.6%)
Surgery duration (mean hours, standard deviation)	5.8 (2.7)
Pelvic fixation	250 (21.3%)
Interbody graft	254 (21.7%)
Any osteotomy	345 (29.4%)
3CO	229 (19.5%)
6–12 posterior levels fused	263 (22.4%)
13+ posterior levels fused	240 (20.5%)
Revision surgery	119 (10.1%)

**Table 2 jcm-11-04436-t002:** Baseline and operative characteristics of all patients stratified by transfusion requirement.

Parameter	No Transfusion	Transfusion	*p*-Value
Age (mean, standard deviation)	54.1 (19.2)	57.2 (18.6)	0.005 *
Sex			
Male	231 (39.4%)	204 (34.8%)	0.107
Female	356 (60.6%)	382 (65.2%)	
ASA Class			
1	38 (6.5%)	18 (3.1%)	<0.001 *
2	244 (41.6%)	176 (30.0%)	
3	294 (50.1%)	372 (63.5%)	
4	11 (1.9%)	20 (3.4%)	
Smoker	102 (17.4%)	73 (12.5%)	0.018 *
Chronic steroid use	23 (3.9%)	33 (5.6%)	0.169
Bleeding disorder	12 (2.0%)	23 (3.9%)	0.058
Dependent functional status	15 (2.6%)	47 (8.0%)	<0.001 *
Body weight (mean kg, standard deviation)	79.3 (20.9)	76.3 (20.9)	0.016 *
Preoperative hematocrit (mean, standard deviation)	41.2 (4.4)	39.9 (4.6)	<0.001 *
Orthopedic surgeon as attending	327 (55.7%)	290 (49.5%)	0.033 *
Surgery duration (mean hours, standard deviation)	4.4 (2.3)	7.1 (2.4)	<0.001 *
Pelvic fixation	43 (7.3%)	207 (35.3%)	<0.001 *
Interbody graft	147 (25.0%)	107 (18.3%)	0.005 *
Any osteotomy	114 (19.4%)	231 (39.4%)	<0.001 *
3CO	63 (10.7%)	166 (28.3%)	<0.001 *
6–12 posterior levels fused	163 (27.8%)	100 (17.1%)	<0.001*
13+ posterior levels fused	63 (10.7%)	177 (30.2%)	<0.001 *
Revision surgery	41 (7.0%)	78 (13.3%)	<0.001 *

* statistically significant result.

**Table 3 jcm-11-04436-t003:** Artificial neural network models’ architectures and accuracy metrics on the testing data.

Parameter	Model 1	Model 2	Model 3	Model 4
Input features	18	18	18	18
Hidden layers	4	4	2	2
Activation function	Sigmoid	ReLU	ReLU	Sigmoid
Accuracy metrics				
Sensitivity	0.79	0.76	0.80	0.71
Positive predictive value	0.72	0.73	0.76	0.75
F1-Score	0.76	0.75	0.78	0.73
Accuracy (ACC)	0.74	0.74	0.77	0.73

## Data Availability

Data are available.

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
