# Peer review of "An Artificial Neural Network Model for the Prediction of Perioperative Blood Transfusion in Adult Spinal Deformity Surgery"

_jcm, 2022, doi:10.3390/jcm11154436_

Round 1

Reviewer 1 Report

This is an excellent study. My minor comments would be:

1. I would recommend explicit use of relevant reporting guidelines (https://www.equator-network.org/reporting-guidelines/guidelines-for-developing-and-reporting-machine-learning-predictive-models-in-biomedical-research-a-multidisciplinary-view/)

2. I think there could be more in reference to existing literature, including those using conventional stats to identify risk factors. 

Reviewer 2 Report

The authors have written an interesting paper on the usage of 'Artificial Neural Network Model for Prediction of Perioperative Blood Transfusion in Adult Spinal Deformity Surgery'. While this is an interesting and well written article, I have the following comments. 
1)It would be useful if the authors elaborate on ANN - the different types of ANN based on architecture (Eg LSTM model, Hopfield network etc) It doesn't have to be in detail but a bit of introduction would help. Especially when the authors themselves have used 2 different models i.e sigmoid and Relu. 
2) An interesting feature is that the model with less hidden layers is performing better. What is the implication of this? Typically Google uses a 30 layered neural network for google photos
3) The review of literature should include comparable studies. Apart from Durand et al, no other equivalent study is reviewed 
